# Association between Cardiovascular Health and Incident Atrial Fibrillation in the General Japanese Population Aged ≥40 Years

**DOI:** 10.3390/nu13093201

**Published:** 2021-09-15

**Authors:** Tetsuo Nishikawa, Yoshihiro Tanaka, Hayato Tada, Toyonobu Tsuda, Takeshi Kato, Soichiro Usui, Kenji Sakata, Kenshi Hayashi, Masa-aki Kawashiri, Atsushi Hashiba, Masayuki Takamura

**Affiliations:** 1Department of Cardiovascular Medicine, Kanazawa University Graduate School of Medical Sciences, Kanazawa 920-8641, Japan; tetsunishi25@gmail.com (T.N.); ttsuda0329@yahoo.co.jp (T.T.); takeshikato@mac.com (T.K.); usuiso@staff.kanazawa-u.ac.jp (S.U.); kenjis@yu.incl.ne.jp (K.S.); kenshi@med.kanazawa-u.ac.jp (K.H.); mk@med.kanazawa-u.ac.jp (M.-a.K.); masayuki.takamura@gmail.com (M.T.); 2Department of Preventive Medicine, Northwestern University Feinberg School of Medicine, Chicago, IL 60611, USA; y.tanaka@northwestern.edu; 3Center for Arrhythmia Research, Northwestern University Feinberg School of Medicine, Chicago, IL 60611, USA; 4Kanazawa Medical Association, Kanazawa 920-0912, Japan; hashiba.atsushi.kanazawa@gmail.com

**Keywords:** cardiovascular health, atrial fibrillation, Japanese

## Abstract

This study explores the association between lifestyle behavior and incident atrial fibrillation (AF) in the general Japanese population. Japanese residents aged ≥40 years undergoing a national health checkup in Kanazawa City were included. We hypothesized that better lifestyle behavior is associated with lower incidence of AF. Lifestyle behavior was evaluated by the total cardiovascular health (CVH) score (0 = poor to 14 = ideal), calculated as the sum of the individual scores on seven modifiable risk factors: smoking status, physical activity, obesity, patterns of eating schedule, blood pressure, total cholesterol, and blood glucose. The association between CVH and incident AF was assessed, adjusting for other factors. A total of 37,523 participants (mean age 72.3 ± 9.6 years, 36.8% men, and mean total CVH score 9 ± 1) were analyzed. During the median follow-up period of 5 years, 703 cases of incident AF were observed. Using a low CVH score as a reference, the upper group (ideal CVH group) had a significantly lower risk of incident AF (hazard ratio [HR] = 0.79, 95% confidence interval 0.65–0.96, *p* = 0.02), especially among those aged <75 years (HR = 0.68, 95% confidence interval 0.49–0.94, *p* = 0.02). Thus, ideal CVH is independently associated with a lower risk for incident AF, particularly in younger Japanese individuals (<75 years).

## 1. Introduction

Atrial fibrillation (AF) is the most common arrhythmia in the world, with an incidence that increases annually, and it is associated with a rising risk of stroke, cardiovascular morbidity, physical disability, dementia, and mortality [1,2,3]. In Japan, the number of patients who will develop AF by 2030 is estimated to be greater than 1 million [4]. Therefore, more studies focusing on preventative approaches to AF are warranted. Established risk factors of AF include aging, hypertension, obesity, smoking, cardiac disease (valvular disease, cardiomyopathy, coronary artery disease, and heart failure), hyperthyroidism, and diabetes mellitus. It is noteworthy that these factors are also known to lead to other cardiovascular diseases [5,6]. Among them, lifestyle behaviors are attracting more attention as modifiable risk factors of AF and other cardiovascular diseases. The American Heart Association already advocates the Life’s Simple 7 (LS7), which consists of seven modifiable lifestyle behaviors and medical factors (diet, obesity, physical activity, smoking status, blood pressure, total cholesterol, and blood glucose) to improve cardiovascular health (CVH) and reduce cardiovascular disease and stroke [7]. Using the LS7 metrics, previous studies revealed that ideal CVH is associated with a reduced risk of AF in Western populations [5,8,9]. Moreover, a cohort study revealed that an intervention for CVH such as reduction in body weight improved arrhythmia-free survival after ablation of AF in the Australian population [10]. However, insufficient data exist regarding this issue in the Asian population, especially among Japanese individuals. Only one study showed abdominal obesity and habitual behaviors, such as smoking status, alcohol intake, and physical activity, to be associated with an increased incidence of AF [11]. Thus, we conducted this study to explore the association between CVH and incident AF in the general Japanese population under the hypothesis that better lifestyle behavior is associated with lower incidence of AF, using large samples (>30,000) of the Japanese-specific health checkups in Kanazawa City.

## 2. Materials and Methods

We included patients who had undergone Japanese-specific health checkups in Kanazawa City, which is a strategy of the Japanese government to provide an early screen for, diagnose, and treat the metabolic syndrome that started in 2008. All general residents in Kanazawa City aged 40 years or older were eligible. The participants completed questionnaires about medical history, medications, and lifestyles. Examinations included anthropometric measurements, physical examinations, blood tests, urine dipstick tests, and resting 12-lead electrocardiogram (ECG).

### 2.1. Study Participants

Eligible participants were Japanese residents aged ≥40 years who had undergone 12-lead ECG at the Japanese-specific health checkups in Kanazawa City in 2013 (*n* = 47,551; Figure 1). We excluded participants with missing baseline characteristics, those who did not complete a follow-up examination at least once during a 5-year follow-up period, those with AF detected at the baseline ECG, and those without adequate follow-up (*n* = 10,028). An event was defined as a new onset of AF diagnosed by automatic analysis of ECG based on the Minnesota code (8-3) during the follow-up period. Results of all automatically coded ECGs were confirmed by experienced physicians for health checkups.

### 2.2. CVH Score

We defined the CVH score to evaluate seven modifiable risk factors following LS7 (Figure 2). The total CVH score ranged from 0 (poor) to 14 (ideal) and was calculated as the sum of the individual scores on seven modifiable risk factors (patterns of eating schedule, obesity, physical activity, smoking status, blood pressure, total cholesterol, and blood glucose). Patterns of eating schedule was scored by three answers from the questionnaire (eating faster than ordinary, eating dinner within 2 h before sleep at least three times per week, and eating snacks after dinner at least three times per week). We scored 2 points (ideal) if answers applied to none of the three questions, 0 points (poor) if the answers applied to all three questions, and 1 point if the answers applied to one or two questions. Obesity was scored by body mass index (BMI). We scored 2 points if BMI was less than 25 kg/m^2^, 0 points if BMI was 30 kg/m^2^ or greater, and 1point if BMI was between 25 and 29.9 kg/m^2^. Physical activity was scored by three answers from the questionnaire (exercising for 30 min per day at least two times per week over 1 year, walking or exercising 1 h per day on a daily basis, walking faster than people of the same sex and age). We scored 2 points (ideal) if the answers applied to all three questions, 0 points (poor) if the answers applied to none of the questions, and 1 point if the answers applied to one or two questions. Smoking status was scored as 2 points for noncurrent smoker and 0 points for current smoker. Blood pressure was scored as 2 points if systolic blood pressure (SBP) was <120 mmHg and diastolic blood pressure (DBP) was <80 mmHg without antihypertensive drugs. It was scored as 0 points if SBP was greater than or equal to 140 mmHg or DBP was greater than or equal to 90 mmHg, and 1 point if it was not applied to either condition. Total cholesterol (TC) was scored as 2 points if the TC was <200 mg/dL without lipid-lowering drugs, 0 points if the TC was greater than or equal to 240 mg/dL, and 1 point if it was not applied to either condition. Blood glucose was scored as 2 points if the fasting blood glucose (FBG) was less than 100 mg/dL without oral hyperglycemic drugs or insulin, 0 points if the FBG was greater than or equal to 126 mg/dL, and 1 point if it was not applied to either condition.

### 2.3. Statistical Analysis

Continuous variables were expressed as mean ± standard deviation, and categorical variables were expressed as number and percentage. Differences in the baseline characteristics were compared using Student’s *t*-test for parametric data and the Mann–Whitney *U* test for nonparametric data. Categorical variables were compared using the chi-square or Fisher’s exact tests. Cox proportional hazard models were used to identify independent associations with the outcomes.

A *p*-value of <0.05 was considered statistically significant. We used R statistical software for all analyses.

### 2.4. Ethical Considerations

The Ethics Committee of Kanazawa Medical Association (16000003) and Kanazawa University (2019-223) approved this study. The research was conducted in accordance with the Declaration of Helsinki (2008) by the World Medical Association. All procedures were performed in accordance with the ethical standards of the responsible committee on human experimentation (institutional and national) and with the Helsinki Declaration of 1975 (as revised in 2008).

## 3. Results

### 3.1. Study Characteristics

Table 1 shows the basic characteristics of this study population. A total of 37,523 participants (mean age 72.3 ± 9.6 years, 36.8% men, and mean total CVH score 9 ± 1) were finally analyzed. During the median follow-up period of 5 years (interquartile range 3.99–5.02), 703 cases of incident AF were observed. There were significant differences in age, sex, SBP, BMI, history of coronary artery disease, stroke, alcohol intake, and estimated glomerular filtration rate (eGFR) between the AF group and the non-AF group. The AF group was significantly older, had a significantly higher proportion of men, and had a significantly greater BMI than the non-AF group. The AF group had a more frequent history of heart disease and stroke. The AF group had a more frequent regular alcohol intake and lower eGFR.

### 3.2. Total CVH Score

Total CVH scores were normally distributed and ranged from 1 to 14, with a mean value of 9.25 ± 1.66. We classified a total CVH score of 1–9 as the poor CVH group (N = 20,177), a total CVH score of 10 as the intermediate CVH group (N = 8819), and a total CVH score of 11–14 as the ideal CVH group (N = 8527), based on the number of individuals according to the distribution. (Figure 3A). We observed 420, 126, and 157 AF incidents among the poor (85,230 person-years), intermediate (37,534 person-years), and ideal (38,349 person-years) groups, respectively (Figure 3B). The incident rate of AF per 1000 was 4.9, 4.1, and 3.6 in the poor, intermediate, and ideal groups, respectively. Compared with the poor CVH group, the ideal CVH group had a significantly lower risk for incident AF (chi-squared test, *p* = 0.0002).

### 3.3. Association between CVH and Incident AF

Using the poor CVH group as a reference, the ideal CVH group had a significantly lower risk of incident AF (hazard ratio (HR) = 0.75, 95% confidence interval 0.61–0.92, *p* = 0.005), in model 1, adjusting for age gender, and regular alcohol intake (Table 2). Likewise, the ideal CVH group had a significantly lower risk of incident AF compared with the poor CVH group (HR = 0.79, 95% confidence interval 0.65–0.96, *p* = 0.02) in model 2, adjusting for age, gender, history of heart disease, history of stroke, alcohol intake, eGFR. In model 2, we also observed other factors that were significantly associated with increased or decreased risk for AF, including age (HR = 1.07, 95% confidence interval 1.06–1.08, *p* = 2.0 × 10^−16^), female sex (HR = 0.48, 95% confidence interval 0.41–0.57, *p* = 2.0 × 10^−16^), no history of heart disease (HR = 0.38, 95% confidence interval 0.32–0.45, *p* = 2.0 × 10^−16^), no history of stroke (HR = 0.78, 95% confidence interval 0.62–0.97, *p* = 0.029), not drinking alcohol (HR = 0.76, 95% confidence interval 0.63–0.92, *p* = 0.005), and eGFR (HR = 0.99, 95% confidence interval 0.989–0.998, *p* = 0.007).

### 3.4. Subanalysis Focusing on the Younger Group (Aged <75 Years)

We also investigated whether the influence of CVH on incident AF was more profound in the younger group as compared with the older group (Table 3). We divided the younger group and elder group by age 75 years based on the following two reasons: (1) age 75 years was close to the median age in this study (Appendix A) and (2) age 75 years or older was defined as advanced elderly in Japan. In participants aged <75 years, using the poor CVH group as a reference, the ideal CVH group had a significantly lower risk of incident AF (HR = 0.64, 95% confidence interval 0.46–0.88, *p* = 0.006) in model 1, adjusting for age, gender, and regular alcohol intake. Likewise, the ideal CVH group had a significantly lower risk of incident AF as compared with the poor CVH group (HR = 0.68, 95% confidence interval 0.49–0.94, *p* = 0.02) in model 2, adjusting for age, sex, history of heart disease, history of stroke, alcohol intake, and eGFR in the younger group. In model 2 in the younger group, we also observed other factors that had a significant difference: age (HR = 1.07, 95% confidence interval 1.04–1.10, *p* = 3.7 × 10^−6^), female sex (HR = 0.42, 95% confidence interval 0.32–0.57, *p* = 5.2 × 10^−9^), no history of heart disease (HR = 0.27, 95% confidence interval 0.20–0.35, *p* = 2.0 × 10^−16^), no history of stroke (HR = 0.54, 95% confidence interval 0.37–0.78, *p* = 0.025), and not drinking alcohol (HR = 0.71, 95% confidence interval 0.53–0.96, *p* = 0.02).

### 3.5. Subanalysis Focusing on the Older Group (Aged ≥75 Years)

In participants aged ≥75 years, there was no significant difference between the poor CVH group and the ideal CVH group (HR = 0.85, 95% confidence interval 0.66–1.10, *p* = 0.21) in model 1, adjusting for age, gender, and regular alcohol intake (Table 4). Similarly, there was no significant difference between the poor CVH group and the ideal CVH group (HR = 0.88, 95% confidence interval 0.69–1.14, *p* = 0.34) in model 2, adjusting for age, sex, history of heart disease, history of stroke, alcohol intake, and eGFR in the older group. In model 2, in the older group, we also observed other factors that were significantly different: age (HR = 1.08, 95% confidence interval 1.06–1.10, *p* = 3.1 × 10^−13^), female sex (HR = 0.52, 95% confidence interval 0.42–0.65, *p* = 2.7 × 10^−9^), no history of heart disease (HR = 0.46, 95% confidence interval 0.38–0.57, *p* = 1.4 × 10^−13^).

## 4. Discussion

Analyzing a large dataset from the Japanese-specific health checkups in Kanazawa City, we observed the following: (1) the ideal CVH was associated with lower incident AF independently of conventional risk factors of AF, (2) an ideal CVH had a larger impact on lowering incident AF in the younger generation (aged <75 years). Our CVH score could be automatically and easily calculated from the questionnaire and measurements obtained from the health checkups. It might be helpful to enlighten participants on their risk of incident AF and encourage the modification of CVH. In observational studies, optimal CVH was associated with a lower risk of incident AF [5,8,9]. In secondary prevention, we observed less frequent AF in the group that had aggressive risk modification, such as with body weight reduction [10]. On the other hand, there were only a few studies regarding this issue in the primary prevention settings [12,13]. Moreover, all of the above studies were from Western countries.

Indeed, ideal CVH is associated with a great reduction in coronary artery disease (79% in men and 72.7% in women), for which the risk factors overlap those of AF [14,15]. Thus, according to these results, as with coronary artery disease, CVH should have a great contribution to incident AF. From the results of our study, a CVH intervention in the younger population might be effective. Therefore, further trials of CVH intervention focused on the younger population are needed. Moreover, we also found that alcohol intake was significantly associated with incident AF as previously described [16]. Accordingly, drinking restrictions should also be considered together with CVH intervention among Japanese as well.

### Limitations

This study has several limitations. First, this was a retrospective study. Second, there were more female participants in the Japanese-specific health checkups in Kanazawa City, which could potentially have affected the results. This is because these health checkups were for housewives or unemployed persons instead of health checkups in their workplace. In Japan, a “regular” worker must undergo health checkups offered by their workplaces, instead of these specific health checkups. Actually, more males work regularly than females in Japan. Third, a diagnosis of AF in the health checkups depended on an ECG that was performed only once per year. Thus, we might have missed paroxysmal AF. Fourth, our definitions of eating habits and exercise were different from those of the American Heart Association’s LS7. For eating habits, our definition focused on eating time and speed of eating. On the other hand, the American Heart Association’s definition focused on the content of the diet. Fifth, this study assessed the participants’ lifestyle at the inclusion cross-sectionally and thus did not address the effect of changes in CVH on incident AF during the follow-up period. Prospective studies with lifestyle interventions are needed to fully address this important issue in the future. Finally, this study did not assess the food composite in these health checkups. However, patterns of eating schedule have been shown to be associated with cardiovascular disease and stroke among the Japanese population [17]; thus, this element is employed in most of the health checkups in Japan. We believe that this factor can serve as a substitute for the food composite, at least among the Japanese population. 

## 5. Conclusions

Ideal CVH is independently associated with a lower risk for incident AF, especially in the younger Japanese population (<75 years).

## Figures and Tables

**Figure 1 nutrients-13-03201-f001:**
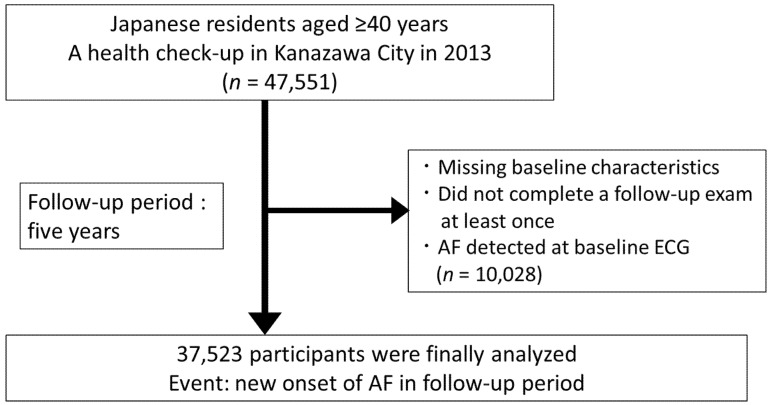
Study flow chart. Eligible participants were Japanese residents aged ≥40 years who had undergone 12-lead electrocardiogram (ECG) during a Japanese-specific health checkup in Kanazawa City in 2013 (*n* = 47,551). We excluded participants with missing baseline characteristics, those who did not complete a follow-up examination at least once during the 5-year follow-up period, those with AF detected at the baseline ECG, and those without adequate follow-up (*n* = 10,028). An event was defined as a new onset of AF diagnosed by automatic analysis of ECG based on the Minnesota code (8-3) during the follow-up period.

**Figure 2 nutrients-13-03201-f002:**
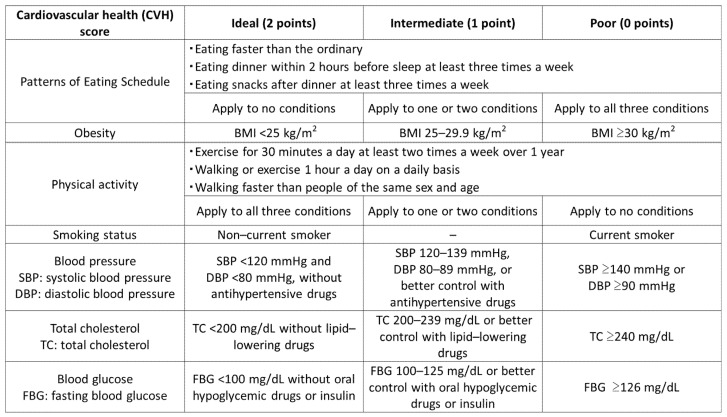
Cardiovascular health (CVH) scoring. The CVH score included seven modifiable components (patterns of eating schedule, obesity, physical activity, smoking status, blood pressure, total cholesterol, and blood glucose). We referred to the data of health checkups from questionnaires, anthropometric measurements, and blood tests. BMI, body mass index.

**Figure 3 nutrients-13-03201-f003:**
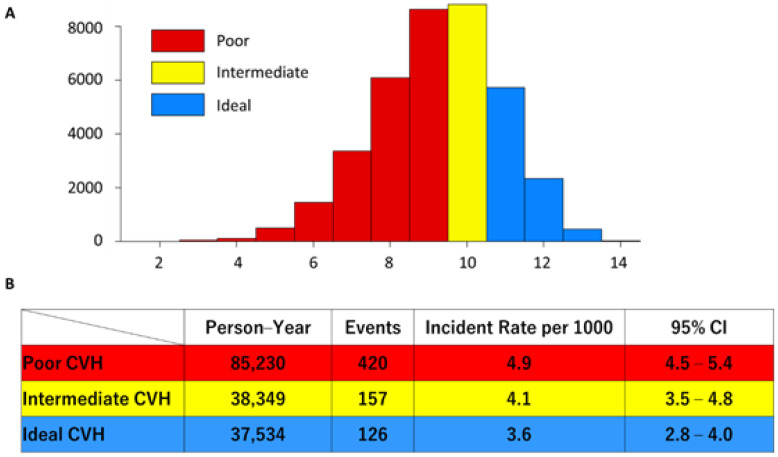
(**A**) Histogram of total cardiovascular health (CVH) score. The horizontal axis shows the total CVH score, and the vertical axis shows the number of participants. We classified a total CVH score of 1–9 as the poor CVH group (red), a total CVH score of 10 as the intermediate CVH group (yellow), and a total CVH score of 11–14 as the ideal CVH group (blue). (**B**) Incident rate of atrial fibrillation by three groups of total CVH scores.

**Table 1 nutrients-13-03201-t001:** Basic characteristics of the study population. Regular alcohol intake meant drinking every day. SBP, systolic blood pressure; DBP, diastolic blood pressure; BMI, body mass index; eGFR, estimated glomerular filtration rate.

Variables	Total	AF (–)	AF (+)	*p*-Value
	N = 37,523	*n* = 36,820	*n* = 703	
Age, years	72.3 (9.6)	72.2 (9.6)	77.3 (8.0)	<0.01
Male, n (%)	13,799 (37%)	13,401 (36%)	398 (57%)	<0.01
SBP, mmHg	128 (15)	128 (15)	130 (15)	<0.01
DBP, mmHg	73 (10)	73 (10)	74 (10)	0.37
BMI, kg/m^2^	22.9 (3.3)	22.8 (3.3)	23.6 (3.4)	<0.01
Smoking, n (%)	3528 (9.4%)	3458 (9.4%)	70 (10%)	0.66
Total cholesterol, mg/dL	196 (33)	196 (33)	185 (31)	<0.01
Fasting blood glucose, mg/dL	104 (29)	104 (29)	108 (31)	<0.01
eGFR, mL/min/1.73 m^2^	71.7 (17.3)	71.8 (17.3)	66.0 (17.2)	<0.01
Coronary artery disease, n (%)	4323 (12%)	4013 (11%)	220 (31%)	<0.01
Stroke, n (%)	2583 (7%)	2494 (7%)	89 (13%)	<0.01
Regular alcohol intake, n (%)	8457 (23%)	8285 (22%)	212 (30%)	<0.01
Total cardiovascular health score	9 (8–10)	9 (8–10)	9 (8–10)	<0.01
Smoking	2 (2–2)	2 (2–2)	2 (1–2)	<0.01
Physical activity	1 (1–1)	1 (1–1)	1 (0–1)	<0.01
Obesity	2 (1–2)	2 (1–2)	2 (1–2)	<0.01
Patterns of eating schedule	1 (1–2)	1 (1–2)	1 (1–2)	<0.01
Blood pressure	1 (1–1)	1 (1–1)	1 (1–1)	0.13
Total cholesterol	1 (1–1)	1 (1–1)	1 (1–2)	0.25
Blood glucose	2 (2–2)	2 (2–2)	2 (2–2)	0.64

**Table 2 nutrients-13-03201-t002:** Association between cardiovascular health (CVH) score and incident atrial fibrillation in all participants. Model 1 was adjusted for age, gender, and regular alcohol intake. Model 2 was adjusted for age, gender, history of heart disease, history of stroke, regular alcohol intake, and estimated glomerular filtration rate (eGFR). The hazard ratio of the intermediate and ideal CVH groups was calculated using the poor CVH group as a reference.

Model 1	Hazard Ratio	Lower 95% CI	Upper 95% CI	*p*-Value
Age	1.08	1.07	1.09	<2 × 10^−16^
Female	0.44	0.37	0.52	<2 × 10^−16^
No alcohol intake	0.82	0.67	0.99	0.04
Intermediate CVH	0.89	0.74	1.07	0.21
Ideal CVH	0.75	0.61	0.92	0.005
Model 2	Hazard ratio	Lower 95% CI	Upper 95% CI	*p*-Value
Age	1.07	1.06	1.08	<2 × 10^−16^
Female	0.48	0.41	0.57	<2 × 10^−16^
No history of heart disease	0.38	0.32	0.45	<2 × 10^−16^
No history of stroke	0.78	0.62	0.97	0.03
No alcohol intake	0.76	0.63	0.92	0.005
eGFR	0.994	0.989	0.998	0.006
Intermediate CVH	0.92	0.76	1.1	0.36
Ideal CVH	0.79	0.65	0.96	0.02

**Table 3 nutrients-13-03201-t003:** Association between cardiovascular health (CVH) score and incident atrial fibrillation in younger participants (<75 years). Model 1 was adjusted for age, gender, and regular alcohol intake. Model 2 was adjusted for age, gender, history of heart disease, history of stroke, regular alcohol intake, and estimated glomerular filtration rate. The hazard ratio of the intermediate and ideal CVH groups was calculated using the poor CVH group as a reference.

Model 1	Hazard Ratio	Lower 95% CI	Upper 95% CI	*p*-Value
Age	1.09	1.06	1.11	1.9 × 10^−9^
Female	0.36	0.27	0.48	1.5 × 10^−12^
No alcohol intake	0.76	0.57	1.03	0.08
Intermediate CVH	0.79	0.58	1.07	0.12
Ideal CVH	0.64	0.46	0.88	0.006
Model 2	Hazard ratio	Lower 95% CI	Upper 95% CI	*p*-Value
Age	1.07	1.04	1.10	3.7 × 10^−6^
Female	0.43	0.32	0.57	5.2 × 10^−9^
No history of heart disease	0.27	0.2	0.35	<2 × 10^−16^
No history of stroke	0.54	0.37	0.78	0.001
No alcohol intake	0.71	0.53	0.96	0.03
eGFR	0.99	0.98	1.00	0.056
Intermediate CVH	0.83	0.61	1.12	0.22
Ideal CVH	0.68	0.49	0.94	0.02

**Table 4 nutrients-13-03201-t004:** Association between cardiovascular health (CVH) score and incident atrial fibrillation in younger participants (≥75 years). Model 1 was adjusted for age, gender, and regular alcohol intake. Model 2 was adjusted for age, gender, history of heart disease, history of stroke, regular alcohol intake, and estimated glomerular filtration rate. The hazard ratio of the intermediate and ideal CVH groups was calculated using the poor CVH group as a reference.

Model 1	Hazard Ratio	Lower 95% CI	Upper 95% CI	*p*-Value
Age	1.09	1.07	1.11	<2.0 × 10^−16^
Female	0.49	0.4	0.61	4.0 × 10^−11^
No alcohol intake	0.84	0.66	1.08	0.19
Intermediate CVH	0.97	0.77	1.22	0.79
Ideal CVH	0.85	0.66	1.10	0.21
Model 2	Hazard ratio	Lower 95% CI	Upper 95% CI	*p*-Value
Age	1.08	1.06	1.10	3.1 × 10^−13^
Female	0.52	0.42	0.65	2.7 × 10^−9^
No history of heart disease	0.46	0.38	0.57	1.4 × 10^−13^
No history of stroke	0.91	0.68	1.21	0.52
No alcohol intake	0.8	0.62	1.02	0.06
eGFR	0.99	0.99	1.00	0.06
Intermediate CVH	0.99	0.79	1.25	0.95
Ideal CVH	0.88	0.69	1.14	0.34

## Data Availability

The data presented in this study are available on request from the corresponding author. The data are not publicly available due to our regulations.

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
