# Peer review of "Association between Cardiovascular Health and Incident Atrial Fibrillation in the General Japanese Population Aged ≥40 Years"

_nutrients, 2021, doi:10.3390/nu13093201_

Round 1

Reviewer 1 Report

General comments:

This manuscript describes the results of a retrospective study considering an association between cardiovascular health (CVH) and atrial fibrillation in the general Japanese population.  The major finding was that ideal CVH was independently associated with lower risk for AF.

The results are not particularly surprising but important.  The manuscript is well written – concise and to the point.

Specific comments:

Abstract:  Please state a hypothesis (same in Introduction).

Line 225:  Insert “with” after “associated”.

Author Response

This manuscript describes the results of a retrospective study considering an association between cardiovascular health (CVH) and atrial fibrillation in the general Japanese population.  The major finding was that ideal CVH was independently associated with lower risk for AF.

 The results are not particularly surprising but important.  The manuscript is well written – concise and to the point.

 Our response: Thank you so much for your great effort on reviewing our manuscript, an giving us favorable comments. We have revised our manuscript according to your comments.

Specific comments:

Abstract:  Please state a hypothesis (same in Introduction).

 Our response: Thank you for your important comment. We added our hypothesis in the abstract and in the Introduction. Please see line 12-13, and 49-50.

Line 225:  Insert “with” after “associated”.

Our response: We revised according to this comment.

Reviewer 2 Report

This is an excellent Japanese study which adds to the existing literature about lifestyle factors and their contribution to incident atrial fibrillation. Some points to consider:

  • The definition of poor, intermediate and ideal CVH were listed in the results section, but would be better served in the methods section. Also, were these arbitrary outpoints (1-9, 10, 11-14), or were these previously defined by the AHA? This was not expressly stated but should be explained further.
  • The higher proportion of females in the study (and the admission in the limitations section that the cohort focused on housewives and unemployed persons) raises the question about whether this study is truly generalizable to the Japanese population as a whole- this should be stated. 
  • One final limitation is the fact that the study does not address the effect of changes in CVH during the 5-year period on incident atrial fibrillation. 

Overall this is an excellent, well written paper that serves as a starting point for Japanese clinicians to address a knowledge gap about lifestyle and medium-term risk of atrial fibrillation. 

Author Response

Reviewer 2

This is an excellent Japanese study which adds to the existing literature about lifestyle factors and their contribution to incident atrial fibrillation. Some points to consider:

Our response: Thank you so much for your great effort on reviewing our manuscript, and giving us useful comments. We have revised our manuscript according to your comments.

  • The definition of poor, intermediate and ideal CVH were listed in the results section, but would be better served in the methods section. Also, were these arbitrary outpoints (1-9, 10, 11-14), or were these previously defined by the AHA? This was not expressly stated but should be explained further.

Our response: Thank you so much for your pointing out an important matter. In fact, the definition of poor, intermediate and ideal CVH have been arbitrary according to the situation, ethnicity, and study population. We defined the threshold based on the number of individuals according to the distribution. We added the descriptions regarding this point together with the numbers of individuals in each category in the text. We believe that this arbitrary nature of this score do not affect our essential message that CVH was associated with incident AF. Please see line 146-149.

  • The higher proportion of females in the study (and the admission in the limitations section that the cohort focused on housewives and unemployed persons) raises the question about whether this study is truly generalizable to the Japanese population as a whole- this should be stated. 

Our response: Thank you so much for your pointing out this important issue. It is true that this health checkups is offered only for individuals without “regular” occupation. In Japan, “regular” worker must undergo health checkups offered by their workplaces, instead of this specific health checkups. Actually, more male are working regularly than female in Japan. So, we fully agree with the reviewer that this is one of the major limitations of this study. We added descriptions regarding this issue as a limitation. Please see line 250-252.

  • One final limitation is the fact that the study does not address the effect of changes in CVH during the 5-year period on incident atrial fibrillation. 

Our response: Thank you again for your pointing out an important issue. We fully agree with the reviewer that this study assessed the participants’ lifestyle at the inclusion cross-sectionally, thus, did not address the effect of changes in CVH on incident AF. We added descriptions regarding this issue as a limitation. Please see line 257-260.

Overall this is an excellent, well written paper that serves as a starting point for Japanese clinicians to address a knowledge gap about lifestyle and medium-term risk of atrial fibrillation.